# A Systematic Review of Complementary and Alternative Veterinary Medicine in Sport and Companion Animals: Soft Tissue Mobilization

**DOI:** 10.3390/ani12111440

**Published:** 2022-06-02

**Authors:** Anna Bergh, Kjell Asplund, Iréne Lund, Anna Boström, Heli Hyytiäinen

**Affiliations:** 1Department of Clinical Sciences, Swedish University of Agricultural Sciences, SE 750 07 Uppsala, Sweden; 2Department of Public Health and Clinical Medicine, Umeå University, SE 901 87 Umeå, Sweden; kjellasplund1@gmail.com; 3Department of Physiology and Pharmacolgy, Karolinska Institutet, SE 171 77 Stockholm, Sweden; irene.lund@ki.se; 4Department of Equine and Small Animal Medicine, Faculty of Veterinary Medicine, University of Helsinki, P.O. Box 57, 00014 Helsinki, Finland; anna.bostrom@helsinki.fi (A.B.); heli.hyytiainen@helsinki.fi (H.H.)

**Keywords:** massage, stretching, myofascial release, soft tissue mobilization, shiatsu, trigger point therapy, tactile therapy, massager machine, massage gun

## Abstract

**Simple Summary:**

Soft tissue mobilization involves different massage and stretching techniques that are commonly used in animals. Despite the frequent use, there is limited knowledge of how the methods affect the animal. Therefore, this study reviews the scientific literature on massage and stretching in cats, dogs, and horses. Three core bibliographic sources were used. Relevant articles were assessed for scientific quality, and information was extracted on study characteristics, species, type of treatment, indication, and treatment effects. Of 1189 unique publications screened, 11 met the inclusion criteria and were included in the review, nine on massage and two on stretching. The risk of bias was assessed as high in eight of the studies and moderate in three of the studies. There were large differences in reported treatment effects; two studies assessed as having a moderate risk of bias indicated a decreased heart rate after massage. According to the results, the scientific documentation of massage and stretching is not of sufficient quality to draw clear conclusions about their clinical effect in horses, dogs, and cats.

**Abstract:**

Soft tissue mobilization is frequently used in the treatment of sport and companion animals. There is, however, uncertainty regarding the efficacy and effectiveness of these methods. Therefore, the aim of this systematic literature review was to assess the evidence for clinical effects of massage and stretching in cats, dogs, and horses. A bibliographic search, restricted to studies in cats, dogs, and horses, was performed on Web of Science Core Collection, CABI, and PubMed. Relevant articles were assessed for scientific quality, and information was extracted on study characteristics, species, type of treatment, indication, and treatment effects. Of 1189 unique publications screened, 11 were eligible for inclusion. The risk of bias was assessed as high in eight of the studies and moderate in three of the studies, two of the latter indicating a decreased heart rate after massage. There was considerable heterogeneity in reported treatment effects. Therefore, the scientific evidence is not strong enough to define the clinical efficacy and effectiveness of massage and stretching in sport and companion animals.

## 1. Introduction

Soft tissue mobilization is an umbrella term for different manual techniques, designed to affect different soft tissue layers, mainly the skin, muscular, fascial, and tendon structures. The manual treatment techniques include for example massage, trigger point therapy, myofascial release, and stretching. Massage has been used since ancient times and is defined as a “manipulation of tissues (as by rubbing, kneading, or tapping) with the hand or an instrument for relaxation or therapeutic purposes [1] or a “mechanical stimulation of soft tissues” [2]. In massage, several techniques can be conducted, ranging from gentle stroking to deep and vigorous kneading of tissues, and techniques can be applied singly or in a mixed combination. The techniques may include effleurage, petrissage, kneading, vibration, tapping, and friction [3]. Stretching involves extending the soft tissues by moving a joint/joints through its normal joint range of motion, separating the origin and insertion of the muscle as far apart as possible. Most soft tissue mobilization techniques are usually applied manually by hand, or with a mechanical device such as a massage gun.

Soft tissue mobilization is applied in order to reduce muscle tension, pain, and stress, as well as to increase blood and lymphatic circulation and elasticity in tissues. Further, it is used in the remodeling of scar tissue [4], with the aim of supporting minimal adhesion formation and the development of optimally functional scar formation. Recently, more emphasis has been put on promoting relaxation and a higher degree of wellbeing through the use of soft tissue mobilization. These effects are described more in terms of “overall relaxation” or “relief of stress”, rather than loosening stiff soft tissues or improving performance [5]. Therefore, the use of soft tissue mobilization may be divided into treatments used for clinical purposes or those with the purpose of promoting wellbeing.

Soft tissue mobilization can be conducted by trained professionals, but it is also performed by laypeople and animal owners. Studies have shown that massage is commonly used both in companion animals and in sport horses [6,7,8,9]. A study on Swiss warmblood horses reports massage to be used in 8% of the horses treated for lameness and back problems [6]. A New Zealand study shows that 26% of dressage riders use different types of massage for their horses and the main indication for using allied health personnel to treat their horse was back problems [7]. The same results are shown in a Swedish study, indicating that massage is predominantly used for back and muscle problems in horses [8]. Further, an international study, asking equine veterinarians what rehabilitation modality they used, stated that 83% of the respondents used stretching and 69% used massage [9]. Despite soft tissue mobilization being frequently used, there is no consensus regarding its clinical effects in animals. To the best of the authors’ knowledge, only a few previous narrative reviews have been published [4,10,11,12]. Therefore, the aim of the present study is to conduct a systematic literature review of soft tissue mobilization therapies. The present article is one of a series of systematic review articles in a Special Issue of Animals on complementary and alternative veterinary medicine (CAVM) therapies used in sport and companion animals. The other articles will cover manipulation/mobilization therapies [13], electrotherapies, therapeutic ultrasound, extracorporeal shock wave therapy, laser therapy, acupuncture, and “miscellaneous therapies” [14].

## 2. Materials and Methods

The overall outline of this systematic review adhered to the Cochrane guidelines on how to perform a systematic review [15], as adapted by the Swedish Agency for Health Technology Assessment and Assessment of Social Services (SBU) in its methodological handbook [16]. 

### 2.1. Review Topic

The aim of the review was to assess the evidence for clinical efficacy and effectiveness of soft tissue mobilization techniques used in sport and companion animals.

### 2.2. Search Strategy

Professional librarians performed searches of Web of Science Core Collection, CABI, and PubMed (1980–2020) in August 2020. The keywords were terms relevant to dog OR cat OR horse, AND veterinary medicine OR veterinary, AND therapy* OR treatment*, AND massage OR stretch OR myofascial OR release OR soft tissue mobilization OR shiatsu OR trigger point OR tactile therapy OR massager machine OR massage gun. 

### 2.3. General Inclusion and Exclusion Criteria

The included studies were to be original research and published in a peer-reviewed journal between the years 1980 and 2020. Primarily observational and interventional studies that would have only one method studied per study group were included. A therapeutic intervention was defined as an intervention intended to reduce the signs, severity, or duration of a clinical condition. Experimental studies, on sound animals, could also be included, when the study would mimic a clinical situation. The subject species were to be canine, feline, or equine, and the full-text version of the article was to be published in English, Swedish, Finnish, German, Italian, Spanish, Portuguese, or French. 

Textbook chapters, conference proceedings, abstracts, opinion notes, review articles, and case reports (subject number <5) were excluded. Experimental studies evaluating mechanisms of action were also excluded. Studies with several methods or multiple treatments simultaneously per intervention group (i.e., massage, transcutaneous electrical stimulation, and non-steroidal anti-inflammatory medication as concomitant interventions for a group) were also excluded. 

### 2.4. Study Selection and Categorisation

All screening was performed based on journal title, publication title, or abstract. Citations identified were imported into Endnote (X9.3.3, 2018) and duplicates were removed. A single author (HH) applied inclusion and exclusion criteria to all publications. 

In the screening phase, articles of possible relevance for the review, namely articles describing one type of intervention in cats, dogs, or horses, were selected for full-text reading. After the first stage of screening, articles deemed potentially relevant were accessed from open access sources. Articles that could not be accessed from digital library resources were requested via the Swedish University of Agricultural Sciences library. For each study, the following key descriptive items were tabulated using templates modified after SBU [16]: first author, year of publication, study design, study population, intervention, type of control, outcome, and relevance (external validity). 

Assessment of the risk of bias (as a measure of scientific quality) of each article was performed in accordance with the Cochrane [15] and SBU [16] guidelines. The assessment was based on the following items: study design, statistical power, deviation from planned therapy, loss to follow-up, type of outcome assessment, and relevance. In the assessment of observational studies, risk of confounding was also included. For consistency, before starting the literature review, three of the authors (KA, HH, ABe) independently screened a random sample of articles, and differences were discussed and resolved before reviewing all articles.

## 3. Results

### 3.1. The Literature Selection Process

In the first phase of the literature selection process, the identification stage, the literature search yielded altogether 1189 references. In the second phase of the process, the screening stage, the references were assessed based on their title and journal-related information, as well as the available abstract. Duplicates and studies with clear breaches of inclusion criteria were excluded at this stage, leaving 25 studies to progress to the eligibility stage of the selection process. In this stage, the full texts of the remaining papers were assessed. After completion of the selection process, 11 articles concerning two therapies, massage and stretching, were retained. The data on the included articles are presented in Table 1 and Table 2, with a description of the article’s content and the level of risk of bias. In total, three randomized controlled trials (RCTs) and eight other types of studies involving a total of 279 horses and 18 dogs were identified.

### 3.2. Massage

Nine studies related to massage therapy with different study designs were found: three RCTs, two randomized cross-over studies, two non-randomized controlled studies, and two cohort studies (Table 1).

#### 3.2.1. Quality of Studies

Six studies had a high risk of bias and three had a moderate risk of bias. No study had a low risk of bias. None of the studies report blinded evaluation. In addition, power or sample size calculations are not mentioned in any of the papers. In most of the randomized papers, the actual randomization process is not described. None of the studies report numbers of animals that were lost to follow-up. Generally, minimal descriptions of the health status of the included animals were provided. This may have resulted in several confounding factors in these studies. For example, in the study of McBride et al. (2004) [23], the lack of exclusion of possible pathologies may have affected the results. An example of another type of confounding factor is the relatively long follow-up time of 7 days after a single massage treatment, with no information on other factors such as treatments or exercise during that time [25]. Yet another issue is the use of healthy animals to study the effect of treatment of pain [25]. Some of the studies use non-validated outcome measures, such as visual movement examination or assessment of racing performance [21,22], which introduces yet another risk for bias.

#### 3.2.2. Clinical Indications

One study evaluated the effect of massage on stereotypic behaviors in horses [24]. The rest of the studies looked at different physiological effects on healthy animals (Table 1).

#### 3.2.3. Interventions and Controls

Various massage techniques were used as interventions in the studies, including an allogrooming type of fingertip massage in certain areas of the back compared with healthy control for the affected group [24] and a specific, pressure on/off type of method with a placebo control group [18]. Effleurage on small areas of the animal at one time was conducted with no control group [23]. Friction, petrissage, shaking, and tapotement were used and compared to randomized control groups [21]. Effleurage and petrissage on proximal body and limbs were compared to non-randomized control groups, one ridden exercise and one non-active group [25]. Friction, petrissage, shaking, and tapotement were compared to a randomized control group in racing horses [22]. Effleurage and kneading of the hindquarters were compared to a randomized cross-over placebo control [19]. A cross-over design was also used to study the effects of petrissage, effleurage, and compression [20]. The effects of Swedish massage (stroking, kneading, stretching) were evaluated after two 40 min sessions once a week for five weeks [17].

#### 3.2.4. Clinical Effects

Seven of the nine studies (five with high risk of bias and two with moderate risk) reported massage having some degree of positive or expected effect, mainly on heart rate and behavior, indicating an overall relaxation effect. One study with a high risk of bias expected the massage to decrease the heart rate of animals with stereotypical behavior, but their results showed the opposite [24]. Another study (with moderate risk of bias) showed no significant change in the creatine phosphokinase (CK) concentration in dogs due to massage [20].

### 3.3. Stretching

Two studies, one non-controlled randomized study and one cohort study, involving a total of 10 dogs and 18 horses were identified (Table 2).

#### 3.3.1. Quality of Studies

One study, with moderate risk of bias, reports a high number of animals having been lost to follow-up. The loss of a few animals has a high impact due to the low original number of animals [26]. The non-randomized controlled trial had a high risk of bias due to a lack of specific description of the randomization as well as the stretching procedure, lack of information related to sample and power calculation, and lack of specific information regarding the horses’ health status [27]. 

#### 3.3.2. Clinical Indications

One study investigated the effect on joint range of motion in dogs with osteoarthritis [26], and the other investigated stride length and range of motion in trot in sound horses [27].

#### 3.3.3. Interventions and Controls

In the canine study, the treatment consisted of 10 s stretches with 10 repetitions of the affected joint, twice a day for 21 days, with no control group [26]. Passive stretching was applied to equine limbs 6 or 3 days a week. Each stretch was performed twice, with 10 + 20 s hold [27]. A control group with no treatment was used. 

#### 3.3.4. Clinical Effects

The canine study shows a significant increase in the range of motion of the affected joints [26]. The equine study reports no significant results in their outcome measures but notes that daily stretching may cause adverse reactions [27].

## 4. Discussion

On the treatment methods reviewed in this paper, it is clear that the amount of evidence is very limited. A larger number of articles is available on massage than on stretching. However, the studies mostly concern general physiological effects rather than the effects on specific clinical indications. However, based on studies of low to moderate quality, the results of the present review suggest that different massage techniques induce a reduction in heart rate and an increase in behavioral signs related to relaxation. These results correspond to studies on humans [28] and in animal models [5], showing some evidence of changes in autonomic tone translated into decreasing heart rate, reducing blood pressure, elevating substances indicative of relaxation such as hormones, and increasing heart rate variability. Interestingly, none of the studies of the two types of methods has examined the effect on muscle perfusion, often used as an explanatory model for its positive effects [3].

As for stretching, the limited number of studies and the diverse results do not enable any conclusions regarding the efficacy of the method. A review of human studies suggests that the immediate effects of stretching are decreased viscoelasticity and increased stretch tolerance, but the effect of stretching over 3 to 4 weeks appears to affect only stretch tolerance, with no change in viscoelasticity [29]. Further, studies from human medicine now speak of a paradigm shift putting more emphasis on active movement rather than passive stretching for increased mobility [30]. 

Several details complicate any conclusions on the included soft tissue mobilization methods. One is the limited number of studies. In addition, the quality of the studies varies considerably, and the studies have a high average risk of bias. Across the board, the outcome variables used in the studies are not equal, and thus the results are not comparable, making their interpretation and use in clinical context challenging. Thus, more research with more unified variable selection related to specific clinical conditions is needed. Generally, the information on the used variables including their operationalization and treatment setting is absent or weak. 

Terminology within soft tissue mobilization is clearly challenging. Primarily, what constitutes massage should be clearly defined in papers describing soft tissue mobilization. Generally, massage is accepted to consist of manual techniques affecting soft tissues. However, one of the papers included in this literature review reports having studied massage, but the technique used actually mimics allogrooming [31]. Thus, the use of the term “massage” in the title of the paper might be misleading. Specificity in the use of terminology related to soft tissue mobilizations is called for. 

Regardless of the soft tissue mobilization technique, it is important to remember that touch in itself might be pleasant and that it offers the potential to discover abnormalities that may otherwise have gone unnoticed. However, when treating illness or injury, the benefits of a treatment must be demonstrated with rigor. 

## 5. Conclusions

Of the two methods included in this review, massage has a higher number of papers reporting on its effects. That being said, many of the papers have a high risk of bias. Moreover, the outcome measures used are very different, as are the massage techniques used. Randomized controlled and blinded trials with sufficient statistical power are needed to gain more understanding of the effects of the techniques on the healthy sport and companion animals as well as on animals with pathologies. 

## Figures and Tables

**Table 1 animals-12-01440-t001:** Summary of studies regarding massage techniques, based on a systematic review of published literature.

Study	Study Design	Control Group	Study Sample	Intervention and Dosage	Outcome Variables	Main Results	Study’s Risk of Bias
Badenhorst, Fourie, Vosloo, 2017 [17]	Randomized cohort	No	10 dressage, 10 endurance, 10 saddlebred horses. Inclusion: - Exclusion: -	Swedish massage (stroking, kneading, stretching): two 40 min sessions once a week for five weeks	Heart rate, AST, CK, flexion and lateral flexion of the neck, height of passive front limb protraction, reach of hind limb in walk	Lower heart rates; improved neck, back, and shoulder range of motion.	High
Birt, Guay, Treiber, Ramirez, Snyder, 2015 [18]	Randomized controlled trial	Yes, placebo	14 quarter horses: 5 control, 9 treatment horses Inclusion: - Exclusion: -	A specific, pressure on/off type of method: 20 min, four times with 9–13 day intervals	Heart rate, surface temperature, behavior	Decrease in heart rate, changes in surface temperature, relaxation-related behavior changes.	Moderate
Hill, Crook, 2010 [19]	Randomized cross-over	Yes, placebo	8 mixed breed horses Inclusion: - Exclusion: skin disease, infection, soft tissue injury, or orthopedic condition	Effleurage and kneading of hindquarters: 30 min, once	Active and passive hind limb protraction	Both active and passive hind protraction increased.	High
Huneycutt, Davis, 2015 [20]	Randomized cross-over	Yes, control	8 minimally conditioned Alaskan husky sled dogs	Massage (petrissage, effleurage, compression): 14 min	Degree of CK release	No significant change in the CK due to massage.	Moderate
Kedzierski, Janczarek, Stachurska, Wilk, 2017 [21]	Randomized controlled trial	Five groups: one “clean” control, two massage, two music	60 3-year-old Arabian horses Inclusion: - Exclusion: -	Relaxing massage (friction, petrissage, shaking, tapotement): either once before official race or every day for 6 months	Heart rate, heart rate variability, cortisol level (saliva), racing performance	Daily massage had more effect than the less frequent one, but both had an effect.	High
Kowalik, Janczarek, Kędzierski, Stachurska, Wilk, 2017 [22]	Randomized controlled trial	Yes, control	72 Arabian horses: 24 control, 48 treatment Inclusion: - Exclusion: -	In specific areas of proximal body, relaxing massage (friction, petrissage, shaking, tapotement): 25 min, 3 days a week, for 1 year	Heart rate and heart rate variability, racing performance	Heart rate and rate variability were positively affected. Massaged horses performed better in races.	Moderate
McBride, Hemmings, Robinson, 2004 [23]	Cohort	No	10 healthy ponies and horses Inclusion: - Exclusion: -	Effleurage on specific areas of proximal body	Heart rate, behavior	Massage of withers and neck decreased the heart rate and caused most positive behavioral responses.	High
Normando, Trevisan, Bonetti, Bono, 2007 [24]	Non-randomized controlled trial	“Clean” control	27 horses: 12 with stereotypic behaviors, 15 without (control) Inclusion: - Exclusion: -	Allogrooming type of fingertip massage along the spine and withers	Heart rate	Massage increased the heart rate of animals with stereotypical behavior and decreased it on the control horses.	High
Sullivan, Hill, Haussler, 2008 [25]	Non-randomized controlled clinical trial	Yes: chiropractic, phenylbutazone, ridden exercise, and no exercise (control)	40 horses without clinical signs of back pain Inclusion: - Exclusion: lameness	Single session of effleurage and petrissage on proximal body and limbs for 35–45 min	Spinal mechanical nociceptive threshold	Mechanical nociception threshold was significantly higher after massage 7 days after treatment.	High

**Table 2 animals-12-01440-t002:** Summary of stretching techniques based on a systematic review of published literature.

Study	Study Design	Control Group	Study Sample	Intervention and Dosage	Outcome Variables	Main Results	Study’s Risk of Bias
Crook et al., 2007 [26]	Cohort	No	10 Labrador retrievers Inclusion: healthy, over 1.5 years old, confirmed osteoarthritis of elbow, stifle, or carpus with limited range of motion Exclusion: Severe pain, other musculoskeletal disorders, long-term corticosteroid treatment, non-compliant owner	10 repetitions of 10 s stretches of the affected joint, twice a day for 21 days	Affected joint range of motion	Significant increase in the range of motion of the affected joints.	Moderate
Rose, Northorp, Brigden, Brigden, Martin, 2009 [27]	Non-randomized controlled trial	Yes, control	18 horses Inclusion: - Exclusion: -	Passive stretching of limbs on 6 or 3 days a week; each stretch performed twice, with 10 + 20 s hold	Stride length and range of motion in trot	No significant increase in stride length: daily stretching may cause adverse reactions.	High

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
