# Peer review of "A Systematic Review of Complementary and Alternative Veterinary Medicine in Sport and Companion Animals: Soft Tissue Mobilization"

_animals, 2022, doi:10.3390/ani12111440_

Round 1
Reviewer 1 Report
General comments
The present investigation has the purpose of carrying out a systematic review of the literature in relation to tissue mobilization therapies in animals. It is an easy manuscript to read and understand for the reader, without great pretensions.
The introduction shows very useful information about the topic to be discussed and presents a clear objective, as well as the reason why to carry out this review. Material and method. Well described. Well-defined inclusion and exclusion criteria. Results well presented and clear and short discussion.
This article is relevant from the point of view of highlighting the research weakness in the use, evaluation of results and clinical knowledge of various soft tissue manipulation therapies, such as massage or stretching.
Format Comments
It would be advisable for authors to always use the same nomenclature or acronyms, for example, in CK or creatine kinase tables.
Main limitation
Unfortunately, the number of articles selected after applying the inclusion and exclusion criteria is very low, only 11, of them 9 on massage and only 2 on stretching. Obviously, this emphasizes the great need for more in-depth research on these therapies in veterinary medicine, but on the other hand, it does not provide much information on these techniques.
Author Response
Thank you for the very positive feedback!! We have revised according to your suggestions regarding the use of the same nomenclature, ie the wording creatine kinase has been changed to CK. Once again, thanks!

Reviewer 2 Report
I do consider the topic VERY relevant to the field - it is the first (to my knowledge) systematic review of this topic. It provides a concise review, also covers the level of bias in the papers available, many of which have high levels of bias. No improvements recommended.
On line 68 - please change "Zeeland" to the correct spelling "Zealand"
On line 154 please change "RTC" to "RCT" as you referenced on line 149
The conclusions and the references are appropriate and relevant No other comments - thank youAuthor Response
Thank you very much for the positive feedback! We have now revised the manuscript according to your suggestions.
On line 68 - please change "Zeeland" to the correct spelling "Zealand"- Done.
On line 154 please change "RTC" to "RCT" as you referenced on line 149- Done.
Once again, thank you for your review!

Reviewer 3 Report
Abstract seems to say nothing can be gleaned. But
Discussion suggests massage does facilitate relaxation
This contradiction needs clarification
Author Response
Thank you for the feedback!
However, there seems to be a part missing regarding the comment on the abstract. The sentence starts with a "But" but then there is nothing more. We have interpret it as the "but" belongs to the sentence on the next line and have tried to revise the manuscript according to your suggestions by adding some results to the simple summary and abstract.
line 23 and 34: the sentence has been changed to "The risk of bias was assessed as high in eight of the studies and moderate in three of the studies, two of the latter indicating a decreased heart rate after massage."
We have also revised the discussion.
line 241: the sentence has been changed to: "However based on studies of low to moderate quality, the results of the present review suggest that different massage techniques induce a reduction in heart rate and increase in behavioural signs related to relaxation. "
Once again, thank you so much for the review!
